# Secretome from Uterine Cervical Mesenchymal Stem Cells as a Protector of Neuronal Cells Against Oxidative Stress and Inflammation

**DOI:** 10.3390/biom15101402

**Published:** 2025-10-02

**Authors:** Javier Mateo, Miguel Ángel Suárez-Suárez, Maria Fraile, Ángel Ramón Piñera-Parrilla, Francisco J. Vizoso, Noemi Eiro

**Affiliations:** 1Department of Orthopaedic Surgery and Traumatology, Hospital Universitario Cabueñes, 33394 Gijón, Spain; negreira84@hotmail.com (J.M.); suarezsmiguel@uniovi.es (M.Á.S.-S.); arpinerap@gmail.com (Á.R.P.-P.); 2Department of Medicine, University of Oviedo, 33006 Oviedo, Spain; 3Research Unit, Fundación Hospital de Jove, 33290 Gijón, Spain; maria.fraile82@gmail.com

**Keywords:** nerve injury, inflammation, oxidative stress, mesenchymal stem cells, secretome, hUCESC, conditioned medium

## Abstract

**Background:** The limited self-repair capacity of nerve tissue requires a new therapeutic approach. Mesenchymal stem cells from the uterine cervix, hUCESC, have shown anti-inflammatory, regenerative, and anti-oxidative stress effects through their secretome, which makes them candidates to evaluate their potential in the context of neuronal damage. In this study, we aimed to determine whether secretome or conditioned medium of hUCESC (hUCESC-CM) has beneficial action in the treatment of PC-12 and HMC3 cells in vitro under conditions of oxidative stress and inflammation. **Methods:** Differentiated PC-12 cells and HMC3 cells were subjected to oxidative stress and inflammatory conditions in the presence of hUCESC-CM. The expression of factors related to both processes was evaluated by q-RT-PCR. **Results:** PC-12 cells treated with hUCESC-CM showed a significant increase in the expression of anti-oxidative stress factors (HO-1 and Nrf2) and a significant decrease in the expression of pro-inflammatory factors (IL1β, IL6 and TNFα). In addition, the treatment of HMC3 cells with hUCESC-CM significantly decreased the expression of IL6 and TNFα and enhanced the expression of neuroprotective factors such as BDNF and GDNF. **Conclusions:** Considering that both oxidative stress and inflammation are interrelated and implicated in several nerve injuries and neurodegenerative disorders, the effects of hUCESC-CM on neuronal cells are very promising.

## 1. Introduction

There are several neurological pathologies that are major public healthcare concerns, as they are a primary cause of death or disability worldwide and can seriously impact the patient’s quality of life. These varied processes include hypoxic–ischemic brain damage in newborns, traumatic brain injury, ischemic and hemorrhagic stroke, degenerative disk disease, Huntington’s disease, Alzheimer’s disease, or Parkinson’s disease. Also, nerve injury can be caused by acute traumatic compression of the nerve in an accident or even by a surgical clamp during which the nerve is not completely transected. Surgically Induced Neuropathic Pain (SNPP) represents a clinical problem, with persistent pain reported in 10–50% of patients following common surgical procedures [1]. Postsurgical neuropathies may be a consequence of transection, contusion, stretching, or inflammation of the nerve [2]. Given the limited intrinsic capacity of nerve tissue to undergo self-repair, there is no effective treatment at present for all these processes, in which oxidative stress and neuroinflammation mediate their occurrences [3].

The brain consumes 20% of the body’s oxygen [4] and produces more reactive oxygen species (ROS) during injuries than other organs. In addition, the brain contains many polyunsaturated fatty acids, which, in addition to their low antioxidant levels, cause the brain to become vulnerable to oxidative stress [5,6]. Regarding lipid peroxidation damage, ROS have diverse dangerous effects, including DNA oxidative damage, protein modifications, activation of protein kinases, activation of redox-sensitive transcription factors, the opening of the ion channels, cell necrosis, apoptosis, and damage to the structure and function of the nervous system [7,8]. H_2_O_2_ is a particularly important ROS [9] because it can cross the cell membrane and produce an apoptotic effect [10]. In addition, oxidative stress can stimulate the inflammatory response. Neuroinflammation is a critical component in the pathogenesis of numerous central nervous system disorders, including Alzheimer’s, Parkinson’s, and other neurodegenerative conditions [11,12]. This process constitutes a complex response to brain injury, characterized by glial activation, release of inflammatory mediators, and generation of reactive oxygen species (ROS) [13,14]. Neuroinflammatory mechanisms also promote the recruitment of circulating macrophages and other peripheral immune cells to the lesion site. Both infiltrating and resident macrophages contribute to the clearance of myelin and axonal debris, a prerequisite for axonal regeneration [15]. Therefore, there is a need for new antioxidant and anti-inflammatory strategies to alleviate oxidative stress and neuroinflammation in the treatment of neurodegenerative diseases and neurological injuries.

Recent studies have confirmed that mesenchymal stem cells (MSC) can promote the repair of nerve injury and improvement of function [16,17,18,19]. Earlier research attributed the therapeutic effects of MSC to their engrafting and differentiation capacity. However, it is currently known that the main effects of MSC are probably mediated by paracrine mechanisms [20]. The secretome, also called conditioned medium, is defined as the set of factors/molecules secreted to the extracellular space. These factors include, among others, soluble proteins, free nucleic acids, lipids, and extracellular vesicles. The latter can be subdivided into apoptotic bodies, microparticles, and exosomes [21]. MSC from the uterine cervix, named hUCESC (human uterine cervical stem cells) have the capacity for self-renewal and differentiation into adipocytes, osteoblasts, and chondrocytes, and the capacity to regulate, through their secretome, several biological processes such as tumor growth, inflammation, oxidative stress, and regeneration [22,23,24,25,26,27]. The novelty of hUCESC-CM compared to conditioned media derived from other MSC sources resides in its origin from human uterine cervical stem cells, which are physiologically adapted to a dynamic microenvironment characterized by constant tissue remodeling, hormonal influence, and exposure to pathogens. These unique biological conditions confer hUCESC with distinctive immunomodulatory and regenerative capacities, resulting in a secretome enriched with factors that display enhanced anti-inflammatory, antimicrobial, and antitumor activities relative to those secreted by MSC isolated from bone marrow, adipose tissue, or the umbilical cord.

In this study, we aimed to determine whether the secretome, also referred to as conditioned medium, of hUCESC (hUCESC-CM) is beneficial in the treatment of PC-12 cells in vitro under conditions of oxidative stress and inflammation.

## 2. Materials and Methods

### 2.1. Ethics Statement

This study adhered to national regulations and was approved by the regional Ethics and Investigation Committee (Comité Ético de Investigación Clínica Regional del Principado de Asturias, ref.: 100/13). All patients provided informed written consent, and human specimens were encoded to protect patient confidentiality.

### 2.2. Isolation and Characterization of hUCESC and Secretome Production

hUCESC were obtained from the cervix tissue (with no abnormalities) of fertile women, as described previously [25]. Briefly, the tissue sample was enzymatically disaggregated and centrifuged for 5 min at 400× *g*, and the pellet was collected and seeded in a culture plate. The sample was cultured in DMEM-F12 with glutamine, penicillin, and streptomycin, 10% FBS, epidermal growth factor (EGF, Gibco, Life Technologies, Paisley, UK), hydrocortisone (Sigma-Aldrich, St. Louis, MO, USA), and insulin (Gibco, Life Technologies), in an air–CO_2_ (95:5) atmosphere at 37 °C. The subculture of cells was carried out with trypsin.

In order to characterize the isolated hUCESC, the cells were stained with a panel of specific monoclonal antibodies for mesenchymal stem cells: CD31-PE, CD34-PE, CD44-PE, CD45-FITC, CD73-PE (Becton Dickinson, Biosciences Pharmingen, San Diego, CA, USA), CD90-FITC, and CD105-FITC (Miltenyi Biotec, Bergisch Gladbach, Germany). Stained cells were analyzed using a CytoFLEX S (Beckman Coulter, Pasadena, CA, USA). The computed data was analyzed using CytExpert 2.1 software provided by the manufacturer.

Secretome from hUCESC (hUCESC-CM) was obtained seeding 1500 cells/cm^2^ and culturing them until they reached 80% confluence. Afterwards, the cells were washed three times in PBS and cultured again in DMEM-F12 without FBS. After 48 h, the medium was centrifuged for 5 min at 300× *g* and the supernatant was collected and then stored at −80 °C until use.

### 2.3. PC-12 and HMC3 Cell Culture

PC-12 cells, a cell line derived from a pheochromocytoma of the rat adrenal medulla, were obtained from the American Type Culture Collection (ATCC, Manassas, VA, USA; CRL-1721). Cells were plated (40,000 cells/well) in a 24-well plate with collagen I (Gibco, Carlsbad, CA, USA; A11428) using DMEM-F12 0.3% BSA, 1X ITS for 24 h. Then, 100 ng/mL NGF was added and the differentiation medium (DMEM-F12 0.3% BSA, 1X ITS with 100 ng/mL NGF) was maintained for 7 days by changing the medium every 2–3 days (without completely removing the medium).

HMC3 cells, a microglial cell line isolated from human brain, were obtained from the American Type Culture Collection (ATCC, Manassas, VA, USA; CRL-3304). Cells were plated (288,000 cells/well) in a 6-well plate using DMEM-F12 with glutamine, penicillin, and streptomycin, with 10% FBS, for 24 h prior induction of inflammation.

### 2.4. Oxidative Stress and Inflammation Induction

For oxidative stress induction, PC-12 cells were washed with PBS and treated for 24 h with 400 μM of hydrogen peroxide (H_2_O_2_) in the presence of DMEM-F12 without FBS and penicillin/streptomycin or in the presence of hUCESC-CM.

For inflammation induction, PC-12 cells were washed with PBS and treated for 18 h with 5 µg/mL LPS in the presence of DMEM-F12 without FBS and penicillin/streptomycin or in the presence of hUCESC-CM. HMC3 cells were washed with PBS and treated for 24 h with IFNγ 10 ng/mL and IL1β 10 ng/mL in the presence of DMEM-F12 with 5% FBS.

Cells treated with DMEM-F12 without FBS and penicillin/streptomycin were used as control. Subsequently, cells were collected and preserved for gene expression analysis.

### 2.5. qRT-PCR

Following stress or pro-inflammatory treatment, a RNeasy Mini Kit (Qiagen, Hilden, Germany) was used for total RNA extraction following the manufacturer’s instructions. The Transcriptor First Strand cDNA Synthesis Kit (Roche, Mannheim, Germany) was used for cDNA synthesis. Reverse transcription was carried out as previously reported [11]. Quantitative real time-PCR (qRT-PCR) was performed using specific primers (PrimeTime primers from IDT, integrated DNA technologies) for the factors studied (BDNF, GDNF, HO-1, Nrf2, IL1β, IL6 and TNFα) and GAPDH as reference gene. Specific primer sequences are shown in Table 1 and Table 2). The mRNA levels were measured in a LightCycler 480 II (Roche) with the following cycling conditions: 95 °C for 10 min, 45 cycles of 95 °C for 10 s, 60 °C for 30 s, and 72 °C for 10 s.

### 2.6. Statistical Analysis

The distribution of variables was assessed using the Kolmogorov–Smirnov test. Data are presented as mean ± standard deviation. Parametric tests, including Student’s *t*-test and one-way ANOVA followed by Bonferroni post hoc comparisons, were used to evaluate differences between groups. Statistical analyses were performed using SPSS version 18.0 (PASW Statistics 18), and a *p*-value ≤ 0.05 was considered statistically significant. All experiments were independently repeated three times to ensure reproducibility.

## 3. Results

### 3.1. Isolation and Characterization of hUCESC

The analysis of hUCESC phenotype by flow cytometry showed that these cells were positive for mesenchymal stem cells markers such as CD44, CD73, CD90, and CD105, while they were negative for CD31 (endothelial marker), CD34, and CD45 (hematopoietic markers) (Figure 1).

### 3.2. PC-12 Cell Differentiation

PC-12 cells were differentiated into neurite-bearing cells and underwent a change in cell body shape from circular to triangular, gradually extending neurites that developed bulbous terminal ends (Figure 2). Morphology was maintained regardless of treatment with or without hUCESC-CM under the different experimental conditions.

### 3.3. Effect of hUCESC-CM on PC-12 Oxidative Stress Injury

In order to study the effects of hUCESC-CM on the expression of antioxidant genes under oxidative stress conditions, we compared the expression levels of heme oxygenase-1 (HO-1) and the nuclear factor erythroid 2 related factor 2 (Nrf2) after H_2_O_2_ treatment in the presence or absence of hUCESC-CM, by qRT-PCR. As shown in Figure 3, mRNA expressions of HO-1 and Nrf2 were significantly increased in cells treated with H_2_O_2_ + hUCESC-CM compared with cells treated with H_2_O_2_ only (*p* < 0.0001).

### 3.4. Effect of hUCESC-CM on Inflamed PC-12

Due to the importance of inflammation in nerve dysfunction, we studied the effect of hUCESC-CM on the expression of pro-inflammatory factors such as interleukin (IL)1β, IL6, and Tumor Necrosis Factor alpha (TNFα). As shown in Figure 4, the expression of IL1β, IL6, and TNFα were significantly increased in PC-12 cells treated with LPS and their expression was significantly decreased in cells treated with LPS + hUCESC-CM, compared with cells treated with LPS only (*p* < 0.01).

### 3.5. Effect of hUCESC-CM on Inflamed HMC3

Due to the impact of inflammation on microglia, we also evaluated the effect of hUCESC-CM on the expression of pro-inflammatory factors such as IL6 and TNFα and on the expression of neuroprotective factors such as BDNF and GDNF. As shown in Figure 5, IL1β + IFNγ induced an overexpression of IL6 and TNFα in HMC3 cells, but the treatment of HMC3 cells with hUCESC-CM significantly decreased the expression of these two inflammatory factors (*p* = 0.038 and *p* = 0.044, respectively). In addition, under pro-inflammatory conditions (treatment of HMC3 cells with IL1β + IFNγ), hUCESC-CM enhanced the expression of neuroprotective factors such as BDNF (*p* = 0.007) and GDNF (*p* = 0.006).

## 4. Discussion

Our results show the potent anti-oxidative stress and anti-inflammatory effects of hUCESC-CM on PC-12 and HMC3 cells, which suggests its interest as a new therapeutic strategy for neurodegenerative diseases and neuronal damage.

Because of the particular nature of the nervous system, it is difficult to culture neurons in vitro. Fortunately, PC-12 is an adrenal pheochromocytoma cell line from rats which, after differentiation, exhibits similarities with sympathetic neurons in morphology and phenotype [28]. This cell line represents a suitable cell line for studying nervous system diseases and is widely used in investigations on neural pathogenesis, neurotoxicity, or neuroprotection, including the study of the oxidative stress damage to cells [29]. In the present study, we used the H_2_O_2_ model as it is considered to be a classical inducer of oxidative stress in living cells [30,31]. H_2_O_2_ is an endogenous ROS which is not only a precursor of hydroxyl free radicals but also possesses signaling capacities [32], and is recognized as a pivotal oxidative stress marker and the main ROS reported in living cells [33].

The generation of ROS is an important characteristic of the initiation of anti-oxidative stress response. Consequently, the expression of the nuclear factor erythroid 2 related factor 2 (Nrf2), a master regulator of anti-oxidative responses, is tightly regulated. Our results show that hUCESC-CM is an effective Nrf2 activator and suggest that it may be a promising candidate for the prevention of neurodegeneration. Nrf2 is a crucial redox-sensitive transcription factor. When cells are exposed to oxidative stress, Nrf2 in the cytoplasm dissociates from keap1 and translocates to the nucleus, activating the expression of more than 200 Nrf2-driven genes that encode antioxidant and detoxification defenses, such as heme oxygenase-1 (HO-1), thioredoxin reductase 1 (TrxR1), NAD(P)H quinone oxidoreductase 1 (NQO1), glutamate–cysteine ligase modifier (GCLM), thioredoxin (Trx1), and glutamate–cysteine ligase catalytic subunit (GCLC), all of which have been studied for their preventive and protective effects against oxidative damage [34]. Accordingly, our data also show that treatment with hUCESC-CM upregulates the expression of HO-1. HO-1 catalyzes heme breakdown to release, among others, biliverdin, which is reduced to bilirubin, a potent radical scavenger. HO-1 provides extensive tissue protection due to its defense mechanism against oxidative stress damage and due to its anti-inflammatory and anti-apoptotic effects. These results suggest that hUCESC-CM enhances the cellular antioxidant response by activating the Nrf2/HO-1 pathway, a key mechanism for protecting neurons against oxidative stress. Indeed, the regulation of Nrf2/HO-1 pathway was proposed as a therapeutic option to slow the progression and ameliorate symptoms of neurodegenerative disorders and neural injuries.

In the present study, we also explored the induction of inflammation in PC-12 cells by LPS, which lead to neuronal injury via facilitating the generations of cytotoxic molecules, such as the pro-inflammatory cytokines IL1β, IL6, and TNFα, as well as chemotactic cytokines [35,36]. We consider that our results demonstrating the inhibitory effect of hUCESC-CM against the gene expression of these cytokines are very relevant. High expression of pro-inflammatory cytokines has been related to several neurodegenerative diseases [37]. IL6 and TNFα, key pro-inflammatory cytokines secreted by activated microglia, contribute to the cytokine cascade during inflammatory response in neuroinflammation [38]. These cytokines are key inflammatory factors involved, for example, in cerebral ischemia and cerebral reperfusion injury. IL6 promote TNFα, which triggers the inflammatory response of tissue and leads to stenosis of local blood vessels. At the same time, IL1β can enhance the adhesion between leukocytes and vascular endothelial cells and the process of leukocytes penetrating the vascular wall, thus stimulating the inflammatory response [39,40]. Therefore, inhibiting the release of IL1β, IL6, and TNFα can effectively reduce neuron apoptosis, improve neurological deficits, reduce the disability rate and improve the quality of life of patients with cerebral ischemia.

In addition, we explored the effect of hUCESC-CM on HMC3 cells, a microglial cell line, under inflammatory conditions, showing a significant decrease in pro-inflammatory factors expression and a significant enhancement of neuroprotective factors. Reducing the expression of pro-inflammatory cytokines such as IL6 and TNFα, while promoting the expression of neurotrophic factors like BDNF and GDNF in microglia, is crucial for mitigating neuroinflammation and supporting neuronal survival. Elevated levels of IL6 and TNFα contribute to neurodegenerative processes by exacerbating inflammatory responses and inducing neuronal damage. Conversely, BDNF and GDNF have been shown to exert anti-inflammatory effects and enhance neuronal resilience. It has been demonstrated that BDNF can inhibit microglial activation and reduce the production of TNFα and IL6. For instance, BDNF can induce dephosphorylation of p38 and JNK, which are critical kinases in regulating inflammatory responses, which leads to the decreased production of IL6 and TNFα in microglia [41,42]. Similarly, GDNF has been found to reduce microglial activation and the associated inflammatory responses. In fact, GDNF can diminish the production of nitric oxide and IL1β, IL6, TNFα, and COX-2 expression in activated microglia by inhibiting the phosphorylation of p38 MAPKs, suggesting that GDNF plays a significant role in modulating neuroinflammation and may contribute to neuroprotection [41,43]. The effects of different sources of MSC and their secretome, including extracellular vesicles (EVs), on human microglial HMC3 cells have been reported. Human adipose tissue-derived (hAD) MSC and their EVs decreased the secretion of pro-inflammatory cytokines such as IL6, IL8, and MCP-1 [44], likewise, bone marrow (BM)MSC-derived exosomes reduced microglial activation and the production of inflammatory markers [45]. These findings suggest that MSC, and particularly their secretome, can modulate the activation and inflammatory response of the microglial, which could have therapeutic implications for neurodegenerative diseases characterized by neuroinflammation.

In this context, it is important to distinguish between the therapeutic implications of administering viable MSC (i.e., via intravenous injection) and the use of MSC-derived secretome (usually called conditioned medium—CM). The administration of MSC introduces living cells that may transiently engraft or become entrapped in specific organs (such as the lungs) and subsequently exert their effects primarily through the secretion of paracrine factors. In contrast, the secretome contains soluble factors and extracellular vesicles secreted by MSC, and thus directly provides the bioactive molecules responsible for most of the therapeutic actions attributed to MSC, without the risks associated with cell survival, biodistribution, or uncontrolled differentiation. Secretome represents a cell-free approach that may circumvent safety concerns and facilitate standardization and storage. These differences highlight complementary but distinct therapeutic strategies. A great advantage of using the secretome of MSC is that the drawbacks of cell therapy can be avoided, including tumor formation, immune rejection, microcirculatory obstruction, arrhythmia, and the transmission of infections [46]. In addition, we can consider other drawbacks than the different administration methods of cell therapies for central nervous diseases. The blood–brain barrier represents a limitation for intravenous injections. At present, the commonly used methods of MSC transplantation are subarachnoid space transplantation or local injection into the injured area [47]. Intrathecal administration requires larger doses because the arachnoid membrane adsorbs a large number of stem cells, which is not conducive to stem cell migration [48]. Intramedullary injections can deliver MSC directly to the site of injury but there is a risk of increased tissue pressure and damage to the normal spinal cord.

The application of the secretome derived from MSC not only circumvents the previously mentioned limitations of cell therapy but also presents additional benefits. For instance, the secretome derived from MSC can be assessed for safety, dosage, and efficacy in a way similar to traditional pharmaceutical products. It can be stored for extended periods without the need for potentially harmful cryopreservatives, maintaining its potency. Utilizing MSC-sourced secretome, like conditioned media, is more cost-effective and practical for clinical use as it eliminates the need for invasive cell collection methods. Additionally, mass production is feasible through customized cell lines in controlled laboratory environments, offering a reliable source of bioactive components. The time and expenses associated with the expansion and upkeep of cultured stem cells can be significantly minimized, allowing for the immediate availability of secretome therapies for urgent conditions such as cerebral ischemia [46].

It should also be noted that the hUCESC secretome contains cytokines with recognized neurotrophic effects, such as VEGF, GM-CSF, SCF, and BNDF [24]. BDNF is a significant neurotrophic factor released by MSC, effectively delaying neuronal death, stimulating neurogenesis, and exhibiting antioxidant effects following ischemic stroke [49,50]. It is predominantly localized within the central nervous system, where it regulates the survival, differentiation, and regeneration of glial cells and neurons; promotes myelination, neuronal migration, and axonal growth; and exerts protective effects on injured neurons [51]. A lack of BDNF may cause neurological conditions such as Parkinson’s disease, brain atrophy, depression, and Alzheimer’s disease [52,53,54]. Oxidative stress may also decrease BDNF expression [55].

Other relevant elements of the hUCESC secretome are the extracellular vesicles [56,57]. Exosomes encapsulate diverse cellular signaling molecules—including proteins, lipids, mRNA, miRNA, lncRNA, circRNA, and DNA—that facilitate intercellular communication through horizontal transfer of these cargos [58,59]. Their therapeutic potential supports the paradigm that tissue regeneration promoted by MSC is largely mediated via paracrine mechanisms, while also offering new opportunities for the advancement of cell-free therapeutic strategies [60]. Functioning as carriers secreted by cells, exosomes are critically involved in the cross-talk among different cell populations [61]. Notably, prior studies have shown that intranasal delivery enables exosomes to reach neurons and microglia through the olfactory and trigeminal nerves, thereby circumventing the blood–brain barrier [62,63,64].

This study has some limitations. First, our analyses were limited to gene expression, without assessing corresponding protein levels; since mRNA and protein abundance do not always correlate, future research should integrate both transcriptomic and proteomic analyses to strengthen the translational relevance of our findings. Second, the immortalized PC-12 cell line was used as a neuronal model, which does not fully represent primary neurons. Third, a more complete characterization of the components of the hUCESC secretome responsible for its neuroprotective effects is still needed. Finally, the analyses were performed without a pharmacological comparator, such as methylprednisolone; although our work focused on exploring the potential of hUCESC-CM as a novel cell-free therapeutic approach, future studies should include such agents. Despite these limitations, our results provide promising evidence of the anti-oxidative stress and anti-inflammatory effects of hUCESC-CM, supporting further investigation of its therapeutic potential in neurological diseases.

## 5. Conclusions

In summary, this study demonstrates that the secretome of hUCESC (hUCESC-CM) exerts significant anti-oxidative and anti-inflammatory effects on PC-12 and HMC3 cells, mediated, at least in part, by activation of the Nrf2/HO-1 pathway and downregulation of pro-inflammatory cytokines. These findings provide molecular insight into the neuroprotective properties of the secretome of hUCESC, reinforcing its potential as a cell-free therapeutic strategy for neurodegenerative diseases and neuronal injury. While further studies are required to validate these effects in vivo and to elucidate the specific bioactive components responsible, our data support the continued development of MSC-derived secretome as a promising alternative to conventional stem cell therapy.

## Figures and Tables

**Figure 1 biomolecules-15-01402-f001:**
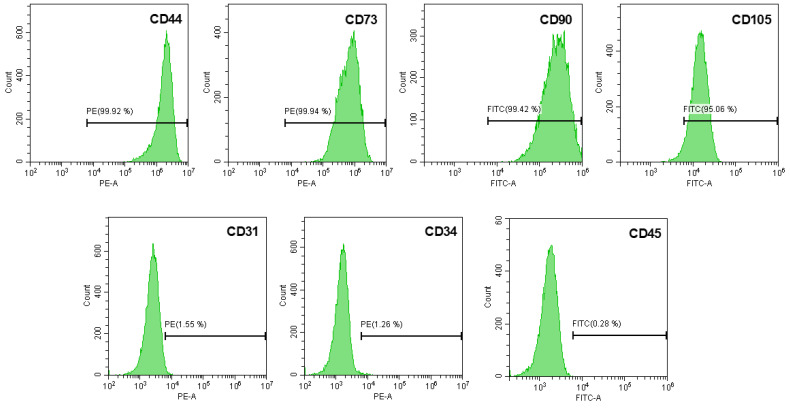
Flow cytometric analysis of hUCESC. Representative figure of hUCESC labeled with FITC- and PE- antibodies and analyzed by flow cytometry. Histograms show the expression of surface antigens.

**Figure 2 biomolecules-15-01402-f002:**
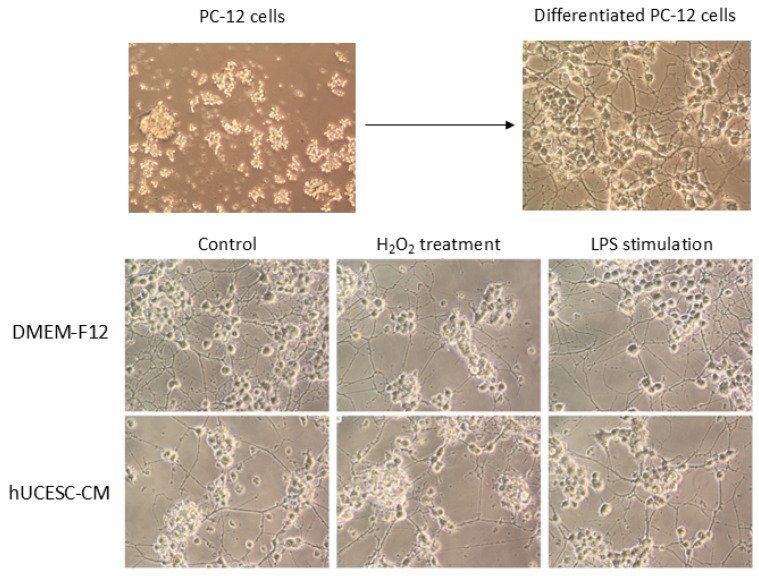
Images of PC-12 cell morphology under different experimental conditions and treatments (with or without hUCESC-CM). 20× magnification.

**Figure 3 biomolecules-15-01402-f003:**
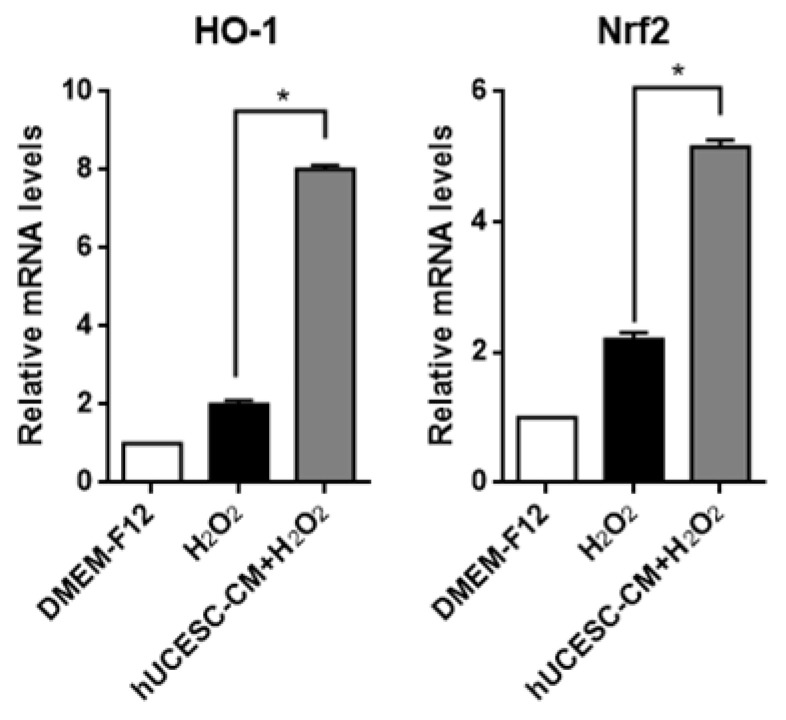
qRT-PCR analysis of HO-1 and Nrf2 in PC-12 cells treated with H_2_O_2_ and PC-12 cells treated with H_2_O_2_ in the presence of hUCESC-CM. Data represent mean ± SD (* *p* ≤ 0.0001).

**Figure 4 biomolecules-15-01402-f004:**
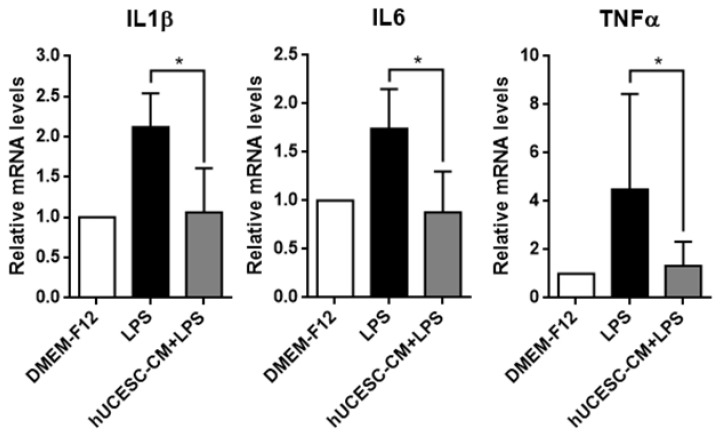
qRT-PCR analysis of IL1β, IL6, and TNFα in PC-12 cells treated with LPS and PC-12 cells treated with LPS in the presence of hUCESC-CM. Data represent mean ± SD (* *p* ≤ 0.01).

**Figure 5 biomolecules-15-01402-f005:**
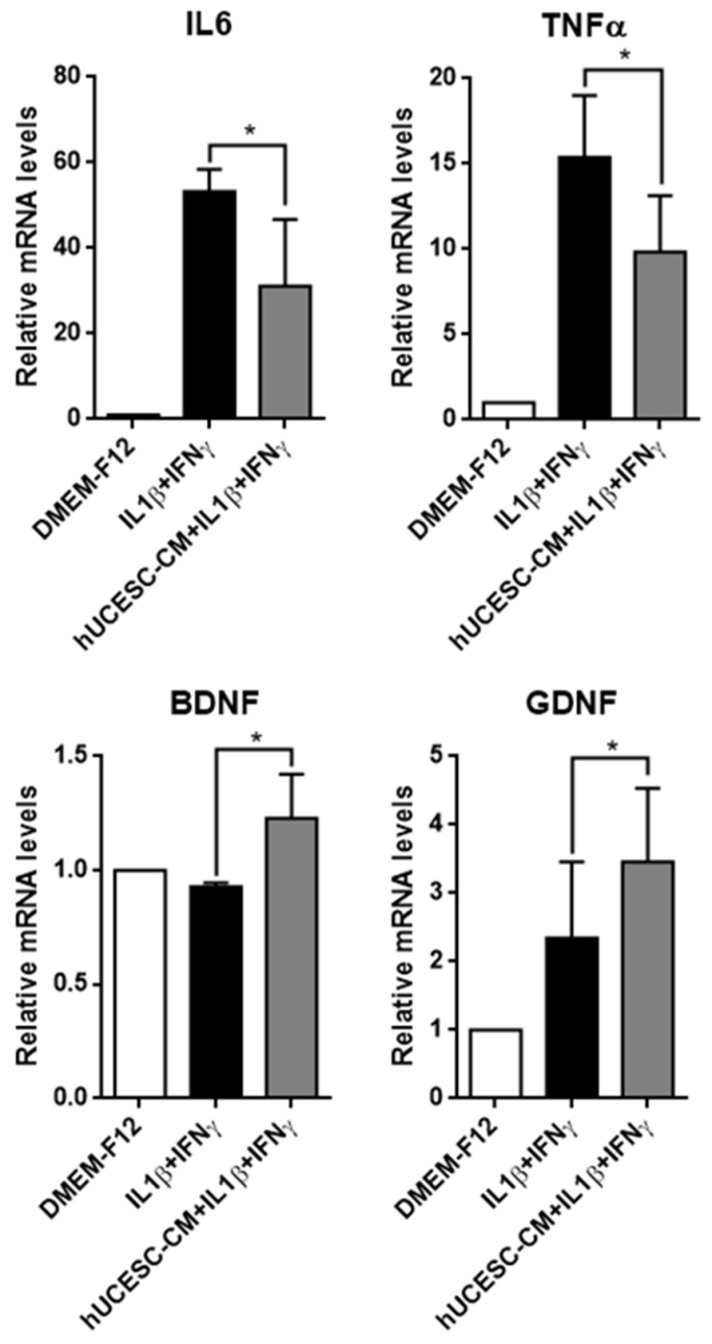
qRT-PCR analysis of IL6 and TNFα and BDNF and GDNF in HMC3 cells treated with IL1β and IFNγ, in the presence or absence of hUCESC-CM. Data represent mean ± SD (* *p* ≤ 0.05).

**Table 1 biomolecules-15-01402-t001:** Rat primer sequences.

Gene	Forward Primer	Reverse Primer	Annealing Temperature (°C)	Product Length (bp)
IL1β	5′CACCTCTCAAGCAGAGCACAG3′	5′GGGTTCCATGGTGAAGTCAAC3′	62.1	79
IL6	5′TCCTACCCCAACTTCCAATGCTC3′	5′TTGGATGGTCTTGGTCCTTAGCC3′	63.9	79
TNFα	5′AAATGGGCTCCCTCTCATCAGTTC3′	5′TCTGCTTGGTGGTTTGCTACGAC3′	63.5	111
HO-1	5′GCCTTCCTGCTCAACATTG3′	5′GCGAAGAAACTCTGTCTGTGA3′	56.1	96
Nrf2	5′CAGTGGATCTGTCAGCTACTC3′	5′CAAGCGACTCATGGTCATCTAC3′	58.9	122
GAPDH	5′AACCCATCACCATCTTCCAG3′	5′CCAGTAGACTCCACGACATAC3′	56.9	85

**Table 2 biomolecules-15-01402-t002:** Human primer sequences.

Gene	Forward Primer	Reverse Primer	Annealing Temperature (°C)	Product Length (bp)
IL6	5′CCTTCCCTGCCCCAGTA3′	5′ATTCGTTCTGAAGAAGAGGTGAGTG3′	59.4	117
TNFα	5′TGCACTTTGGAGTGATCGG3′	5′TCAGCTTGAGGGTTTGCTAC3′	56.7	145
BDNF	5′AATGCTCACACTCCACATCC3′	5′CAAACATAGGTCCTTCCGTCA3′	56.7	100
GDNF	5′GGCACCTGGAGTTAATGTCC3′	5′CCACGACATCCCATAACTTCA3′	58	98
GAPDH	5′ACATCGCTCAGACACCATG3′	5′TGTAGTTGAGGTCAATGAAGGG3′	56.4	143

## Data Availability

The original contributions presented in the study are included in the article; further inquiries can be directed to the corresponding author.

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
