# Peer review of "Secretome from Uterine Cervical Mesenchymal Stem Cells as a Protector of Neuronal Cells Against Oxidative Stress and Inflammation"

_biomolecules, 2025, doi:10.3390/biom15101402_

Round 1

Reviewer 1 Report

Comments and Suggestions for Authors

The manuscript titled “Mesenchymal stem cells secretome from uterine cervix as a protector of neuronal cells against oxidative stress and inflammation” addresses a relevant and interesting topic. The paper is well written in English and the discussion is overall appropriate. However, several important issues need to be addressed before the manuscript can be considered for publication. Please find below my detailed comments:

  1. Experimental support for the conclusions
  • The study relies mainly on gene expression analysis of a limited number of inflammatory and/or oxidative stress markers. This is not sufficient to support the conclusions.
  • Additional experiments at the protein expression level are strongly recommended, such as immunofluorescence, Western blot, and ELISA assays.
  1. Conditioned medium (CM) characterization
  • Please specify the total number of cells used to obtain the conditioned medium.
  • It would be advisable to test CM derived from tissues of at least three different donors (n ≥ 3) to ensure reproducibility and biological variability.
  1. Figure presentation and data reproducibility
  • In Figure 3, error bars are missing. If only one experiment was performed, this is not sufficient: the experiments should be repeated to provide statistical robustness.
  • In all figure legends, please indicate the number of independent experiments performed.
  1. Discussion
  • I suggest adding a statement regarding the future applications that the authors would propose based on their findings.
  • It would also be interesting to test and discuss the effect of CM treatment for longer exposure times.

I consider that additional experimental work is necessary to reinforce the findings. In particular, validation at the protein level is essential before the manuscript can be considered for publication.

Author Response

Please specify the total number of cells used to obtain the conditioned medium.

Secretome from hUCESC (hUCESC-CM) was obtained seeding 1,500 cells/cm2 and culturing them until they reached 80% confluence”, as indicated in the revised version of the manuscript (Material and methods section, line 117). The production is carried out under standardized conditions based on a technology transfer has been done to a CDMO for the GMP production for a further clinical trial.

It would be advisable to test CM derived from tissues of at least three different donors (n ≥ 3) to ensure reproducibility and biological variability.

The homogeneity of hUCESC-CM among donors has been previously established, setting selection criteria and in process controls (IPC) during production and for the hUCESC-CM validation, as is done for the GMP production.

In Figure 3, error bars are missing. If only one experiment was performed, this is not sufficient: the experiments should be repeated to provide statistical robustness.

The error bars were added to the revised version of the manuscript; it was an oversight.

Please indicate the number of independent experiments performed.

In the revised version of the manuscript, we have indicated in the Statistic analysis that “All experiments were independently repeated three times to ensure reproducibility” (line 168).

I suggest adding a statement regarding the future applications that the authors would propose based on their findings.

We thank the reviewer for this suggestion. While our findings are promising, we believe that proposing specific future applications at this stage would be too hypothetical. Therefore, we have included a statement in the discussion highlighting that further studies are required to deepen the understanding of the mechanisms and therapeutic potential of hUCESC-CM before considering clinical or practical applications.

It would also be interesting to test and discuss the effect of CM treatment for longer exposure times.

We thank the reviewer for this insightful comment. We agree that evaluating the effects of hUCESC-CM over longer exposure times would be very interesting. Although it was beyond the scope of the present study, we have acknowledged in the discussion that future investigations should explore the sustained neuroprotective effects of hUCESC-CM.

Additional experiments at the protein expression level are strongly recommended, such as immunofluorescence, Western blot, and ELISA assays.

We thank the reviewer for this valuable observation, we note that a similar suggestion was also made by another reviewer, and following both recommendations, we included this point as a limitation of the study in the revised version of the manuscript (line 357):

This study has some limitations. First, our analyses were limited to gene expression, without assessing corresponding protein levels; since mRNA and protein abundance do not always correlate, future research should integrate both transcriptomic and proteomic analyses to strengthen the translational relevance of our findings (…). Despite these limitations, our results provide promising evidence of the anti-oxidative stress and anti-inflammatory effects of hUCESC-CM, supporting further investigation of its therapeutic potential in neurological diseases.

Reviewer 2 Report

Comments and Suggestions for Authors
  1. Overall Assessment

This paper describes the neuroprotective effect of cervical MSC derived secretome. This is a well-structured and relevant experimental study addressing the neuroprotective potential of conditioned medium from human uterine cervical stem cells (hUCESC-CM) on neuronal and microglial cells under oxidative and inflammatory stress. The objective was evaluate hUCESC-conditioned medium (CM) for PC-12 neuronal cells (oxidative/inflammatory stress) and HMC3 microglia (inflammatory stress). They explained the mechanisms via hUCESC-CM activates Nrf2 pathway, suppresses cytokine expression and enhances neurotrophic factors.

They concluded that hUCEC-CM offers protection via antioxidant with anti-inflammatory effect and it can be used a strong candidate for cell-free therapy in neurodegeneration. MSCs have antioxidant and anti-inflammatory effect via paracrine effect.

The work is original, methodologically sound, and timely, particularly in the context of regenerative medicine and cell-free therapies. However, several grammatical issues, unclear sentences, and minor inconsistencies in the use of scientific terminology need revision before acceptance.

📋 Detailed Comments

  1. Title and Abstract
  • ✅ Title is clear and informative.
  • ❗ Abstract has minor grammatical issues:
    • Example: “which make them candidates to evaluate…” → should be “which makes them candidates for evaluation…”
    • Missing space in "GDNF.Conclusion:" → should be "GDNF. Conclusion:"
  1. Introduction
  • ✅ Strong scientific rationale provided.
  • ❗ Language issues:
    • "due them are primary cause" → should be “as they are a primary cause…”
    • "These variate processes include…" → “These varied processes include…”
    • "degenerative disc disease, Huntington´s disease, Alzheimer’s disease or Parkinson" → consistent disease naming preferred: “Parkinson’s disease”
  • 💡 Suggestion: Clearly define the novelty of hUCESC-CM compared to other MSC sources earlier.
  1. Methods
  • ✅ Methodological approach is comprehensive and clearly described.
  • ❗ Some inconsistencies:
    • “store at -80ºC until used” → should be “stored at -80ºC until use”
    • Repetition of “were obtained from ATCC” can be made more concise.
  • 💡 Suggestion: Include specific catalog numbers for cell lines for reproducibility.
    • The limitation of this study is only in vitro study, for better understanding, in vivo study is necessary.
  1. Results
  • ✅ Good figures and logical result progression.
  • ❗ Repeated grammatical issues:
    • “the expression was significantly decreased…” → should be “their expression was significantly decreased…”
    • “under pro-inflammatory conditions… hUCESC-CM enhanced the expression…” → rephrase to avoid awkward clause chaining.

  1. Discussion
  • ✅ Very comprehensive with good mechanistic interpretation.
  • ❗ Needs proofreading:
    • “Because of the particularly of nervous system…” → “Because of the particular nature of the nervous system…”
    • “we also explore the induction…” → tense inconsistency; use “explored”
    • “using MSC-sourced secretome, such as CM, is more economical…” → awkward phrasing; consider rephrasing to “The use of MSC-sourced secretome (e.g., CM) is more economical…”
  • 💡 Suggestion: Separate long paragraphs for better readability.
    • Please add description the difference between MSC cell therapy via IV injection and conditioned medium in the DISCUSSION section.
  •  
  1. Figures and Legends
  • ✅ Adequate and labeled correctly.
  • ❗ Grammatical errors in legends:
    • “Data represents SD (*p ≤ 0.01).” → should be “Data represent mean ± SD…”
  1. References
  • ✅ Appropriately cited and formatted for MDPI style.
  • ❗ Minor issue: Ensure uniformity in journal abbreviations (e.g., “Mol Neurodegener” vs. “Mol Neurodegeneration”).

Authors conclude that hUCEC-CM offers neuroprotection through antioxidant with antiinflammtory effect as cell-free theraphy in neurodegeneration. The drug control is necessary such as methylprednisolone, it contains antioxidant and anti-inflammatory effect. Please compare hUCEC-CM vs methylprednisolone.

✍️ Grammar and Language Corrections (Selected Examples)

Original Sentence

Corrected Version

“due them are primary cause of death…”

“as they are a primary cause of death…”

“These variate processes include…”

“These varied processes include…”

“Because of the particularly of nervous system…”

“Due to the unique characteristics of the nervous system…”

“cells were positive for mesenchymal stem cells markers…”

“cells were positive for mesenchymal stem cell markers…”

“store at -80ºC until used”

“stored at -80ºC until use”

“we also explore the induction…”

“we also explored the induction…”

“Data represents SD”

“Data represent SD”

“under conditions of oxidative stress and/or inflammation”

“under conditions of oxidative stress and inflammation” (remove “/or” for clarity)

Recommendation

Minor Revisions Required – The study is scientifically sound and highly relevant to the field of regenerative neurology. Before publication, the authors should address the grammatical inconsistencies, improve clarity and flow in some paragraphs, and ensure precise scientific phrasing throughout.

Author Response

This paper describes the neuroprotective effect of cervical MSC derived secretome. This is a well-structured and relevant experimental study addressing the neuroprotective potential of conditioned medium from human uterine cervical stem cells (hUCESC-CM) on neuronal and microglial cells under oxidative and inflammatory stress. The objective was evaluate hUCESC-conditioned medium (CM) for PC-12 neuronal cells (oxidative/inflammatory stress) and HMC3 microglia (inflammatory stress). They explained the mechanisms via hUCESC-CM activates Nrf2 pathway, suppresses cytokine expression and enhances neurotrophic factors.

They concluded that hUCEC-CM offers protection via antioxidant with anti-inflammatory effect and it can be used a strong candidate for cell-free therapy in neurodegeneration. MSCs have antioxidant and anti-inflammatory effect via paracrine effect.

The work is original, methodologically sound, and timely, particularly in the context of regenerative medicine and cell-free therapies. However, several grammatical issues, unclear sentences, and minor inconsistencies in the use of scientific terminology need revision before acceptance.

We are grateful to the reviewer for his/her comments.

Grammatical issues

We have corrected the text according to the suggestions.

Introduction. Strong scientific rationale provided. Clearly define the novelty of hUCESC-CM compared to other MSC sources earlier.

Following the recommendation, we define the novelty in the revised version of the manuscript: “The novelty of hUCESC-CM compared to conditioned media derived from other MSC sources resides in its origin from human uterine cervical stem cells, which are physiologically adapted to a dynamic microenvironment characterized by constant tissue remodeling, hormonal influence, and exposure to pathogens. These unique biological conditions confer hUCESC with distinctive immunomodulatory and regenerative capacities, resulting in a secretome enriched with factors that display enhanced anti-inflammatory, antimicrobial, and antitumor activities relative to those secreted by MSC isolated from bone marrow, adipose tissue, or umbilical cord.” (Introduction section, line 84).

Discussion. Very comprehensive with good mechanistic interpretation. Please add description the difference between MSC cell therapy via IV injection and conditioned medium in the DISCUSSION section

In the revised version, we have expanded the Discussion section to explicitly describe the differences between mesenchymal stem cell (MSC) therapy via intravenous injection and the use of MSC-derived conditioned medium.

In this context, it is important to distinguish between the therapeutic implications of administering viable MSC via intravenous injection and the use of MSC-derived secretome (usally called conditioned medium - CM). Intravenous delivery of MSC introduces living cells that may transiently engraft or become entrapped in specific organs (such as the lungs), and subsequently exert their effects primarily through the secretion of paracrine factors. In contrast, secretome contains soluble factors and extracellular vesicles secreted by MSC, and thus directly provides the bioactive molecules responsible for most of the therapeutic actions attributed to MSC, without the risks associated with cell survival, biodistribution, or uncontrolled differentiation. CM represents a cell-free approach that may circumvent safety concerns and facilitate standardization and storage. These differences highlight complementary but distinct therapeutic strategies.” (Discussion section, line 303)

Authors conclude that hUCESC-CM offers neuroprotection through antioxidant with antiinflammtory effect as cell-free theraphy in neurodegeneration. The drug control is necessary such as methylprednisolone, it contains antioxidant and anti-inflammatory effect. Please compare hUCESC-CM vs methylprednisolone.

We appreciate the reviewer’s valuable suggestion regarding the inclusion of drug control such as methylprednisolone, which indeed has well-described antioxidant and anti-inflammatory properties. However, the present study was specifically designed to investigate the potential of hUCESC-CM as a novel cell-free therapy. While a direct comparison with methylprednisolone would certainly provide further insights, it was beyond the scope and feasibility of the current work. We fully agree that future studies should incorporate methylprednisolone or other pharmacological controls to establish a more direct comparison between hUCESC-CM and clinically used anti-inflammatory/antioxidant drugs. We have now acknowledged this point in the discussion section as a limitation and a relevant direction for future research (line 357).

This study has some limitations. (…) the analyses were performed without a pharmacological comparator, such as methylprednisolone; although our work focused on exploring the potential of hUCESC-CM as a novel cell-free therapeutic approach, future studies should include such agents. Despite these limitations, our results provide promising evidence of the anti-oxidative stress and anti-inflammatory effects of hUCESC-CM, supporting further investigation of its therapeutic potential in neurological diseases.

Minor Revisions Required – The study is scientifically sound and highly relevant to the field of regenerative neurology. Before publication, the authors should address the grammatical inconsistencies, improve clarity and flow in some paragraphs, and ensure precise scientific phrasing throughout.

We sincerely thank the reviewer for the positive evaluation of our study and for the constructive comments. We have carefully revised the manuscript to address grammatical inconsistencies, improve clarity and flow in the text, and ensure precise scientific phrasing throughout. We believe these changes have enhanced the readability and overall quality of the manuscript.

Reviewer 3 Report

Comments and Suggestions for Authors

The manuscript by Javier Mateo and colleagues is devoted to the study of the neuroprotective activity of the secretome of cervical MSCs. The manuscript has an average level of novelty, since many neuroprotective properties of MSCs from other organs and tissues have already been demonstrated previously. 

There are serious comments on the work itself, which don't let accept it in its current form:
1. The authors do not mention the novelty of their work. This must be added to the Abstract and Introduction!
2. The authors do not explain why they use MSCs from the uterine cervix, named hUCESC. Why these cells? Why this source of MSCs?
3. The authors characterize hUCESC, but it is necessary to indicate this in the section “Isolation of hUCESC and conditioned medium production”. For example, like this: "The characterization of the obtained cells was carried out by staining with MSC-specific antibodies following by flow cytometry measurement, as described below." Otherwise, at this stage, the question arises: were the cells characterized?
4. There is no sequence, annealing temperature, and amplicon length of primers used.
5. The section “Materials and Methods” lacks a statistical analysis section. It is unclear how the statistical analysis was performed.
6. In general, there is no information in the paper about the number of replicates in each experiment. This information should be indicated in the Materials and Methods section when describing the methodology, under the figures, and when describing the results next to "p<..."
7. There is no standard deviation in Fig. 3. If this was a single measurement, then statistical data processing cannot be performed.
8. Lines 209-210: "Our results show that hUCESC-CM is an effective Nrf2 activator and suggest that it may be a promising candidate for the prevention of neurodegeneration." Where is this shown? This is the first mention of this protein in the study. If the authors mean the Nrf2/HO-1 axis, the importance of the Nrf2 protein and its connection with HO-1 must be indicated above, where the HO-1 expression level is assessed.
9. One of the serious limitations of this study is that only the RNA level is assessed without measuring the protein level. The levels of proteins and mRNAs do not always correlate. This must be noted as the limitation of the study!!!
10. The unit of measurement on the 0Y axis in Figures 3 and 4 raises a question. Is it a multiple coefficient or a percentage? Why is the expression level in the control group equal to 100? 
11. In the text, the authors use the terms “secretome” and “conditioned medium” interchangeably. One of them should be chosen and used throughout the manuscript.

Comments on the Quality of English Language

A serious drawback of the manuscript is the presence of a large number of complicated sentence constructions and incorrect use of some words. This makes the manuscript difficult to read. I strongly recommend the manuscript be read and corrected by a colleague whose native language is English. Below there are just few examples of such phrases and word usage:
1. "Mesenchymal stem cells secretome from uterine cervix" - part of the title. It follows that the secretome was obtained from the cervix, not from cells obtained from the cervix. The title needs to be corrected!!!
2. "Because of the particularly of nervous system" - incorrect use of the word “particularly”
3. It is not clear who or what: "it can effectively reduce neuron apoptosis" (line 240)
4. "BDNF can avoid phosphorylation of p38 and JNK" (line 253) - does BDNF really avoid phosphorylation of p38 and JNK?
5. "A great advantage of using the secretome of MSC is that the drawbacks of cell therapy can be avoided" - a very complex sentence construction
6. etc., etc., etc.

Author Response

The authors do not mention the novelty of their work. This must be added to the Abstract and Introduction! The authors do not explain why they use MSCs from the uterine cervix, named hUCESC. Why these cells? Why this source of MSCs?

We note that a similar suggestion was also made by another reviewer, and following both recommendations, we have now explicitly highlighted the novelty of our work in the revised version. Specifically, we added a paragraph in the Introduction section to clarify how our study contributes new insights into the field.

The novelty of hUCESC-CM compared to conditioned media derived from other MSC sources resides in its origin from human uterine cervical stem cells, which are physiologically adapted to a dynamic microenvironment characterized by constant tissue remodeling, hormonal influence, and exposure to pathogens. These unique biological conditions confer hUCESC with distinctive immunomodulatory and regenerative capacities, resulting in a secretome enriched with factors that display enhanced anti-inflammatory, antimicrobial, and antitumor activities relative to those secreted by MSC isolated from bone marrow, adipose tissue, or umbilical cord.” (Introduction section, line 84).

The authors characterize hUCESC, but it is necessary to indicate this in the section “Isolation of hUCESC and conditioned medium production”. For example, like this: "The characterization of the obtained cells was carried out by staining with MSC-specific antibodies following by flow cytometry measurement, as described below." Otherwise, at this stage, the question arises: were the cells characterized?

The characterization was described in a paragraph which has been transferred to the “Isolation and characterization of hUCESC and conditioned medium production” section in the revised version of the manuscript (line 110):

hUCESC were stained with a panel of specific monoclonal antibodies for mesenchymal stem cells: CD31-PE, CD34-PE, CD44-PE, CD45-FITC, CD73-PE (Becton Dickinson, Biosciences Pharmingen, San Diego, CA, USA), CD90-FITC, CD105-FITC (Miltenyi Biotec, Bergisch Gladbach, Germany). Stained cells were analyzed using a CytoFLEX S (Beckman Coulter). The computed data was analyzed using CXP software provided by the manufacturer.

There is no sequence, annealing temperature, and amplicon length of primers used.

The information has been included in the revised version of the manuscript (Material and methods section, Table 1 and 2).

The section “Materials and Methods” lacks a statistical analysis section. It is unclear how the statistical analysis was performed.

We thank the reviewer for pointing this out. The omission of the statistical analysis section in the original manuscript was an oversight. We have now added a detailed description in the “Materials and Methods” section (line 165).

The distribution of variables was assessed using the Kolmogorov-Smirnov test. Data are presented as mean ± standard deviation. Parametric tests, including the Student’s t-test and one-way ANOVA followed by Bonferroni post-hoc comparisons, were used to evaluate differences between groups. Statistical analyses were performed using SPSS version 18.0 (PASW Statistics 18), and a p-value ≤ 0.05 was considered statistically significant. All experiments were independently repeated three times to ensure reproducibility.

In general, there is no information in the paper about the number of replicates in each experiment. This information should be indicated in the Materials and Methods section when describing the methodology, under the figures, and when describing the results next to "p<..."

We apologize for the oversight in the original manuscript. We have now included the number of experimental replicates in the Materials and Methods section, (line 163): “All experiments were independently repeated three times to ensure reproducibility.”

There is no standard deviation in Fig. 3. If this was a single measurement, then statistical data processing cannot be performed.

The error bars were added; it was an oversight.

Lines 209-210: "Our results show that hUCESC-CM is an effective Nrf2 activator and suggest that it may be a promising candidate for the prevention of neurodegeneration." Where is this shown? This is the first mention of this protein in the study. If the authors mean the Nrf2/HO-1 axis, the importance of the Nrf2 protein and its connection with HO-1 must be indicated above, where the HO-1 expression level is assessed.

We thank the reviewer for this important comment. To clarify, the activation of Nrf2 by hUCESC-CM was assessed together with HO-1 expression under oxidative stress conditions. Specifically, as described in the Results section, we measured mRNA levels of both Nrf2 and HO-1 by qRT-PCR after H₂O₂ treatment in the presence or absence of hUCESC-CM. Our data (Figure 3) show that co-treatment with hUCESC-CM significantly increased the expression of both HO-1 and Nrf2 compared with H₂O₂ alone (p<0.0001), supporting the conclusion that hUCESC-CM acts as an Nrf2 activator. We have now clarified in the text the connection between Nrf2 and HO-1 and the relevance of this axis in the antioxidant response (line 256).

One of the serious limitations of this study is that only the RNA level is assessed without measuring the protein level. The levels of proteins and mRNAs do not always correlate. This must be noted as the limitation of the study!!!

We thank the reviewer for this valuable observation. Following the recommendation, we included this point in the revised version of the manuscript (line 357):

This study has some limitations. First, our analyses were limited to gene expression, without assessing corresponding protein levels; since mRNA and protein abundance do not always correlate, future research should integrate both transcriptomic and proteomic analyses to strengthen the translational relevance of our findings. (…). Despite these limitations, our results provide promising evidence of the anti-oxidative stress and anti-inflammatory effects of hUCESC-CM, supporting further investigation of its therapeutic potential in neurological diseases.

  1. The unit of measurement on the 0Y axis in Figures 3 and 4 raises a question. Is it a multiple coefficient or a percentage? Why is the expression level in the control group equal to 100?

In the revised version of the manuscript, the Y-axis in Figures 3 and 4 has been clarified: the expression levels are now normalized to the control group, which is set as 1. This approach ensures a clear and consistent representation of relative gene expression levels, avoiding confusion with percentages or multiple coefficients.

  1. In the text, the authors use the terms “secretome” and “conditioned medium” interchangeably. One of them should be chosen and used throughout the manuscript.

We have chosen to use the term secretome throughout the text to describe the biological material. However, for the figures, we have retained the abbreviation hUCESC-CM also to maintain consistency with previous studies of the group. We believe this approach preserves both clarity and continuity with established literature.

Round 2

Reviewer 1 Report

Comments and Suggestions for Authors

The authors have appropriately replied to my comments and, having also acknowledged the limitations of their work, I consider the manuscript suitable for publication.

Author Response

Reviewer 1

The authors have appropriately replied to my comments and, having also acknowledged the limitations of their work, I consider the manuscript suitable for publication.

We are grateful to the reviewer for all his/her comments which undoubtedly result in an improved version of the article.

Reviewer 3 Report

Comments and Suggestions for Authors

The authors have significantly improved the manuscript and made revisions to it in accordance with the comments.
There are few minor recommendations regarding the manuscript and the design of the experiment:
1. in the section "Ethics statement" provide, please, the number of the protocol, that allows your team to work with human cells
2. I recommend to include the explanation for what the cells "were stained with a panel of specific monoclonal antibodies".
For example, you can write it like this: "In order to characterize the isolated hUCESC these cells were stained with a panel of specific monoclonal antibodies"
3. is it right? - 72ºC for 1 sec (line 152)?
4. there are still no annealing temperatures and PCR-products' lengths for each pair of the primers
5. I've checked several primers (rats, IL1b, IL6) and found that they belong to the same exon. In future design the primers that anneal the adjacent exons, not the same one - to avoid the possible non-specific qPCR signal from genomic DNA.

Author Response

Reviewer 3

In the section "Ethics statement" provide, please, the number of the protocol, that allows your team to work with human cells

We have accordingly modified this section as follows: “This study adhered to national regulations and was approved by the regional Ethics and Investigation Committee (Comité Ético de Investigación Clínica Regional del Principado de Asturias, ref.: 100/13). All patients provided informed written consent All and human specimens were encoded to protect patient confidentiality.” (line 106).

I recommend to include the explanation for what the cells "were stained with a panel of specific monoclonal antibodies". For example, you can write it like this: "In order to characterize the isolated hUCESC these cells were stained with a panel of specific monoclonal antibodies"

Following the recommendation, we have modified the paragraph as follows: “In order to characterize the isolated hUCESC, the cells were stained with a panel of specific monoclonal antibodies for mesenchymal stem cells: CD31-PE, CD34-PE, CD44-PE, CD45-FITC, CD73-PE (Becton Dickinson, Biosciences Pharmingen, San Diego, CA, USA), CD90-FITC, CD105-FITC (Miltenyi Biotec, Bergisch Gladbach, Germany). Stained cells were analyzed using a CytoFLEX S (Beckman Coulter). The computed data was analyzed using CXP software provided by the manufacturer.” (line 121).

Is it right? - 72ºC for 1 sec (line 152)?

We thank the reviewer for pointing this out. We have corrected the mistake: 72ºC for 10 sec. (line 164).

There are still no annealing temperatures and PCR-products' lengths for each pair of the primers.

We have added annealing temperatures and PCR-products' lengths for each pair of the primers in the new Table 1: Rat primer sequences. (line 166).

I've checked several primers (rats, IL1b, IL6) and found that they belong to the same exon. In future design the primers that anneal the adjacent exons, not the same one - to avoid the possible non-specific qPCR signal from genomic DNA.

Thank you for your observation and helpful suggestion. For this study, we ensured DNase treatment of RNA samples. Nevertheless, we agree that designing primers spanning exon–exon junctions or across adjacent exons would provide an additional layer of specificity, and we will incorporate this strategy in future primer designs to enhance the accuracy of our qPCR results.